# Recent Progress of SERS Nanoprobe for pH Detecting and Its Application in Biological Imaging

**DOI:** 10.3390/bios11080282

**Published:** 2021-08-19

**Authors:** Lei Zhang, Qianqian Zhao, Zhitao Jiang, Jingjing Shen, Weibing Wu, Xingfen Liu, Quli Fan, Wei Huang

**Affiliations:** 1Institute of Advanced Materials (IAM), Nanjing University of Posts and Telecommunications (NUPT), 9 Wenyuan Road, Nanjing 210023, China; m18860438668@163.com (Q.Z.); jiangzhitaoxy@163.com (Z.J.); iamjjshen@njupt.edu.cn (J.S.); iamxfliu@njupt.edu.cn (X.L.); iamqlfan@njupt.edu.cn (Q.F.); iamwhuang@njupt.edu.cn (W.H.); 2Jiangsu Provincial Key Lab of Pulp & Paper Science & Technology, Nanjing Forestry University, 159 Longpan Road, Nanjing 210023, China; wbwu@njfu.edu.cn; 3Frontiers Science Center for Flexible Electronics (FSCFE), Shaanxi Institute of Flexible Electronics (SIFE), Northwestern Polytechnical University (NPU), 127 West Youyi Road, Xi’an 710072, China; 4Frontiers Science Center for Flexible Electronics (FSCFE), Shaanxi Institute of Biomedical Materials and Engineering (SIBME), Northwestern Polytechnical University (NPU), 127 West Youyi Road, Xi’an 710072, China

**Keywords:** pH, SERS, probe, detection, imaging

## Abstract

As pH value almost affects the function of cells and organisms in all aspects, in biology, biochemical and many other research fields, it is necessary to apply simple, intuitive, sensitive, stable detection of pH and base characteristics inside and outside the cell. Therefore, many research groups have explored the design and application of pH probes based on surface enhanced Raman scattering (SERS). In this review article, we discussed the basic theoretical background of explaining the working mechanism of pH SERS sensors, and also briefly described the significance of cell pH measurement, and simply classified and summarized the factors that affected the performance of pH SERS probes. Some applications of pH probes based on surface enhanced Raman scattering in intracellular and extracellular pH imaging and the combination of other analytical detection techniques are described. Finally, the development prospect of this field is presented.

## 1. Introduction

It is well known that many life activities in cells are accompanied by changes in pH, so local detection of pH in cells is important to understand the mechanism of their activities. Mitochondrial respiration and fermentation metabolism produce large amounts of CO_2_ and lactic acid, respectively. All tissues produce acid, including tumors. Compared with other ions, the detection and quantification of hydrogen ions in living cells are of more concern in the field of biosensors at present, because they play a crucial role in the physiological and pathological processes of single cells and organisms, and the change of their concentrations directly affects the physiological functions of normal living organisms [1]. This information is essential for a better understanding of a wide range of physiological and metabolic processes as well as some biotechnological applications. Surface enhanced Raman scattering has been widely used in many fields due to its unique advantages such as high sensitivity, strong specificity, good reuse ability and optical stability, especially in biological detection and biological imaging in recent years [2,3,4]. Compared with other pH detection imaging technologies, the pH probe based on SERS mainly has the following advantages: (1) there is no damage to the detected biological samples; (2) the fluorescence of biological system and the Raman signal of water can be ignored; (3) there will be no quenching or photobleaching. Therefore, the construction of pH SERS probe with good performance is of great significance for the study of physiological activities and related biochemical reactions in cells, which also provides a powerful platform for the pathological study of many diseases.

In this article, the latest progress in the design, development and application of cell pH probes based on surface enhanced Raman scattering spectroscopy in detecting various biological systems is reviewed from the aspects of pH Raman signal molecules and sensor substrates. The motivation of measuring cell pH and the mechanism of pH SERS detection was also briefly introduced. In principle, the pH measurement based on SERS mainly relies on the measurement of the corresponding spectral changes of the pH sensor, such as the change of peak height and peak position, etc. Unlike the direct determination of many other compounds in cells, the determination of pH in cells is usually based on the measurement and analysis of the relationship between the various types of spectral characteristic peaks of the signal molecules. The most important aspect is to select appropriate pH Raman signal molecules, which will respond to the pH and base conditions of the environment and provide the relevant SERS spectrum. Therefore, it is necessary to make a systematic summary and classification of Raman signal molecules required to construct pH SERS probes. Secondly, the structure and assembly design of pH SERS probes is also an important factor affecting the sensitivity and reliability of the sensor, so it is necessary to review its design strategy. Finally, some recent hot research reports on SERS probes for pH imaging in and out of living cells were also summarized. We hope that this review will encourage more researchers to join the research field and provide guidance for new research teams to conduct research in this field.

## 2. The Significance of pH SERS Probe in the Study of Physiological Activities of Cells

### 2.1. Surface Enhanced Raman Scattering (SERS)

Raman scattering is a very weak phenomenon, but it has been found that the surface enhancement Raman scattering effect of organic molecules adsorbed on the surface of plasmonic nanoparticles (gold, silver, etc.) can increase the original very weak Raman signal by up to 14 orders of magnitude [5], which makes it possible to construct chemical sensors based on the inherent chemical specificity and very low detection limit of Raman spectroscopy. It was first discovered in the mid-1970s by Fleischmann [6] and his colleagues when they measured the Raman scattering of pyridine on a coarse silver electrode, and they attributed this enhancement to the surface area effect with an enhancement factor of 10^5^–10^6^ [7]. Subsequently, Moskovits [8] proposed that the SERS strength was related to the enhanced field excited by local surface plasma in the nanostructured metal. Surface enhanced Raman scattering spectroscopy is a noninvasive photon scattering technique that provides molecular specific information about the vibrational energy levels of molecules. It is a unique vibrational mode based on the target analyte to provide specific chemical information, and has been widely used in a variety of biomedical detection and imaging [9]. Electromagnetic enhancement mechanism (EM) and chemical enhancement mechanism (CM) are the two main mechanisms of SERS enhancement. The former is mainly related to the resonance excitation of the surface body of metal nanoparticles, while the latter involves the photoinduced charge transfer transition between substrate Fermi level and molecular energy level. As shown in Figure 1, the first three mechanisms (Figure 1a–c) belong to chemical enhancement mechanism, and the latter (Figure 1d) belongs to electromagnetic enhancement mechanism [10]. Among them, the improvement of Raman signal in SERS effect mainly comes from the electric field intensity when the plasma nanostructure is irradiated by laser, that is, physical enhancement [11].

The enhancement of the Raman signal is mainly due to the surface plasmon effect of the nanosubstrate material. The local surface plasmon resonance excited under the action of incident light greatly enhances the electromagnetic field around the substrate, which enhances the Raman signal of the object to be measured. The space area with significant enhancement effect is called “hotspot” area, which are junctions, interstices, and tips of metallic nanostructures. The construction of substrate materials with high sensitivity, good stability and easy regulation of “hot spot” was the key to SERS technology research. Raman imaging technology is a new generation of fast, high precision, surface scanning laser Raman technology, which is a perfect combination of microscope technology and laser Raman spectroscopy technology. It has the characteristics of high-speed, noninvasive, label-free and extremely high resolution imaging. During Raman imaging, we scanned the measured area of the sample with a laser with a preset wavelength and obtained the Raman spectrum of each set point and then the Raman image with highly precise structural and chemical information is obtained from the chemometric analyzed Raman spectra [12]. In general, particles with large size and tip morphology (such as nanoflowers, nanostars, and so on) or in an aggregated state will have stronger Raman signals [13,14].

### 2.2. Significance of Cell pH Value in the Study of Cell Physiological Activities

Cell pH differences mainly comes from the H^+^ and biological chemical water ionization. The concentration of H^+^ can affect most aspects of cell function [1], such as cell metabolism [15] (the change of the enzyme active site shape, even make its degeneration inactivation), cytoskeleton elements [16], (actin and tubulin, etc.) crosslinking, aggregation, proliferation and migration, the ability of muscle cells to generate tension [17] and the working efficiency of ion selective channels [18]. The pH value of body fluids, organs, different cells and tissues in the organism is usually strictly regulated by acid–base homeostasis, such as lysosome, endoplasmic reticulum (ER), Golgi apparatus, lipid droplets and other organelles. The regulation and stability of intracellular pH value has a significant impact on a series of physiological activities of cells. In addition, the change of pH value can also change the charge properties of the substrate, so that the substrate can neither bind to the active site nor catalyze [19]. Talley and colleagues were the first to achieve in situ pH values in cells using a 4-mercaptobenzoic acid functionalized silver nanospheres pH probe [5], recognizing that acid–base reaction molecules within cells are highly dependent on external stimuli [20]. Changes in pH value can also be caused by some cellular or tissue lesions, such as epilepsy, Parkinson’s disease, and Alzheimer’s disease [21]. Similarly, the pH of cells is not uniform, and different organelles within the same cell have different pH values. For example, the pH value of mitochondria is about 8.0. The pH value of cytoplasm is basically maintained at about 7.2, and the acidity of lysosome is relatively high, about 4.5–5.0. The cell needs to stabilize these values through a series of biochemical reactions and physiological activities. Therefore, it is of great significance to realize the real-time monitoring of the pH of cell tissue for understanding the physiological activities of organisms, and even to study the working mechanism of some diseased cells and tissues.

## 3. pH SERS Probe Molecules

In recent years, surface enhanced Raman scattering has shown a very high application prospect in pH detection of microenvironment, and it is also of great significance to understand various biological processes and multiphase chemical reactions. We know that the reason why pH SERS probes could have a certain response to the acidity and alkalescence of the environment was mainly because the pH probe molecules had a certain pH response. To put it simply, pH probe molecules are mainly a class of substances that are very sensitive to changes in the pH of the surrounding environment [22]. They are functional groups with a series of protonation and deprotonation reactions. When the pH of the external environment changes, their Raman scattering spectra will change accordingly with it. At present, the sensitive molecules mainly used to construct pH SERS probes are some carboxyl, amino and pyridine rings containing three kinds of functional groups with pH response. Among them, the three most representative signal molecules are 4-mercaptobenzoic acid (4-MBA) [23,24,25], 4-mercaptopyridine (4-Mpy) [26,27,28], 4-aminobenzene mercaptan (4-ATP) [19,29,30], etc. pH sensitive molecules are usually bound together by the formation of sulfhydryl groups and amino groups with precious metals. For these mainstream pH-sensitive molecules, Huang et al. [29] systematically evaluated the pH response range, anti-co-existing ions, accuracy and stability of the pH SERS probes constructed by them. Among them, they demonstrated that 4-MBA exhibited good sensing performance in neutral and alkaline regions, while the pH response range of 4-Mpy and 4-ATP tended to be acidic and neutral. It is worth mentioning that the pH probe constructed using 2-mercaptobenzoic acid (2-ABT) [30] had poor performance. Therefore, this paper mainly discusses the action mechanism and application research of the above three common pH-sensitive molecules with good performance in detail, and will not repeat too much about 2-ABT.

### 3.1. 4-Mercaptobenzoic Acid

Among the three kinds of mainstream pH probe molecules, 4-mercaptobenzoic acid was often widely used as pH Raman report molecules by various research groups to construct pH SERS probes [31,32]. The main reasons why it can be used as a Raman report molecule for the construction of pH Raman probe are its simple structure, wide pH response range, high photochemical stability, mercaptan mediated metal surface affinity and its ability to form a single molecular layer on the metal surface through molecular self-assembly. As shown in Figure 2a, the reason why 4-MBA could be used for the construction of pH SERS probe was mainly because its -COO^−^ tensile vibration peak located around 1420 cm^−1^ increased with the increase of pH value. When pH was greater than ~6.5, the increase of de-proton carboxyl group led to a larger increase, and sometimes the peak value would also be redshifted with the increase of pH value. This suggests that –COO^−^ may be involved in the formation of hydrogen bonds. On the contrary, the strength of the symmetric tensile vibration peak at -C=O at 1690 cm^−1^ will decrease with the decrease of pH [33]. This opposite trend also makes 4-MBA widely used in the field of pH SERS probes. Park et al. [25] coated 4-mercaptobenzoic acid functionalized gold nanorods on microneedle nanorods made from a commercial polymer binder and found that the pH probe had a good linear relationship between Raman signal and pH value in the pH range of 5~9. It can also be used for in situ pH detection of human skin (Figure 2d). Xie et al. [23] based on gold nanoflower as SERS substrate loading with 4-MBA to develop a new SERS nanoprobe for pH mapping, which can be used to distinguish normal cells and tumor cells. Monica et al. [24] fixed 4-mercaptobenzoic acid onto the surface of chitosan-coated silver nanotriangles to generate a robust pH SERS traceable system. These experiments show that 4-mercaptobenzoic acid is indeed a very potential pH-sensitive molecule, and the research on its application in the construction of pH SERS probes has been very mature and extensive.

### 3.2. 4-Mercaptopyridine

In addition to the 4-mercaptobenzoic acid mentioned above, 4-mercaptopyridine is also one of the pH reporter molecules often used by research groups in this field to construct pH SERS probes. The main mechanism of its pH response is that the N in the pyridine ring contains a lone pair of electrons, which can bind to H^+^ in the environment under acidic conditions (Figure 2b), so it can respond to the pH value in the environment to a certain extent. Brandon et al. [26] 4-Mpy coated gold nanoparticles (AuNPs) undergo aggregation resulting from N−AuNP bond formation in both acidic and basic environments. Bai et al. [27] combined SERS and etching technology to develop a pH SERS nanosensor with 4-mercaptopyridine as probe molecule, and coated its surface with bovine serum protein (BSA) to improve the stability and biocompatibility of nanoparticles. Shen et al. [28] developed and designed a gold nanorod (AuNRs) functionalized by a pH-responsive molecule (4-mercaptopyridine, 4-Mpy) and a peptide that can specifically deliver AuNRs to a target subcellular organelle to determine the pH value of a specific organelle. As shown in Figure 2e, Xu’s research group also use the 4-Mpy as the pH reporter and to develop a pH SERS probe with linear response within a pH value from 4 to 8.5 covers the full range of the possible pH values of the extracellular environment [34]. It can be seen from the above report that 4-Mpy is also a good pH responsive Raman reporter molecule.

### 3.3. 4-Aminobenzene Mercaptan

The pH-dependent surface enhanced Raman scattering of 4-aminobenzene mercaptan on metal nanoparticles has been proved by a large number of studies, but its mechanism is still not very clear. We believe that with the decrease of pH value, the amino group in 4-aminobenzene mercaptan tends to be protonated, and conversely, it tends to form a pair double bond (Figure 2c). This will make the corresponding Raman scattering spectrum change accordingly so as to achieve the purpose of pH detection. Ji et al. [35] analyzed the pH-dependent behavior of 4-ATP in a static buffer solution with pH from 3.0 to 2.0, and studied the change sequence of different vibration intensity by using two-dimensional correlation SERS spectrum. They suggest that the amino protonation of 4-ATP in an acidic medium leads to the rearrangement of the electron clouds in the benzene ring, thereby altering the vibration of the ring skeleton. Zong et al. [19] reported a pH SERS probe treated with 4-ATP functionalized hydrochloric acid (Figure 2f), which realized the detection of intracellular pH value. Yang et al. [36] also used 4-ATP to modify a new type of composite nanofiber for the construction of pH SERS probe. Chen’s research group [37] reported a 4-ATP functionalized multi-walled nanotubes coated with silver coated gold nanoparticles nanospheres as pH SERS probes, which had a wider pH response range (3.0–14.0), good stability and biocompatibility.

### 3.4. Other Novel pH Probe Molecules

In addition to the above several commonly used pH probe molecules, many interesting new pH probe molecules have been studied and discussed by many research groups in recent years. For example, Lawson [39] reported a novel pH-sensitive disulfide reporter molecule. The difference between 2,5-dimercapto benzoic acid and 4-mercapto benzoic acid is that the benzene ring of this molecule contains two sulfhydryl groups, which makes it easy to join two metal nanoparticles together to form a dimer. The probe molecule acts as a pH-sensitive molecule and also acts as a dimer bridging molecule. In recent years, Paulo et al. [40] calculated the characterized vibrational assignment of the spectrum of thionicotinamide (TNA), thioisonicotinamide (iTNA) and 5-(4-pyridyl)-1,3,4-oxadiazole-2-thiol (Hpyt) on gold surface and characterized the self-assembled monolayers (SAMs). Although structurally similar, they exhibit different chemical behaviors in the same acid–base environment. At pH = 6, the self-assembled monolayer of Hpyt on gold surface is not protonated, while TNA and iTNA are partially protonated and completely protonated, respectively. In the process of developing other excellent pH reporting molecules, Kong et al. [41] designed and utilized a new type of pH composite reporting molecule to develop a pH sensing technology based on SERS. Interestingly, they used arene chromium tricarbonyl linked aminothiophenol (Cr(CO)_3_-ATP) to design novel pH-sensitive molecules. The mechanism of pH-sensing is mainly the protonation and deprotonation of the pH-responsive amino group in ATP, which causes the internal electronic changes at the coupling point between the aromatic ring and Cr(CO)_3_. This is reflected in the Raman peak movement of the -CO tensile vibration ~1820 cm^−1^, which breaks the traditional method of constructing the probe by analyzing the signal strength of the traditional pH Raman probe.

## 4. pH SERS Probe

Surface enhanced Raman scattering shows a very high prospect in biomedical detection and biological imaging. The unique optical properties and chemical stability of noble metal nanoparticles make them ideal probes for studying biological systems. For example, silver nanoparticles are used as substrates due to their strong local surface plasmonic resonance (LSPR) properties. Compared with the former, gold nanoparticles have better chemical stability and lower biological toxicity [42]. However, the SERS performance of metal nanoparticles depended on the size, shape and structure of the nanoparticles besides the influence of the material type. In this study, Zhang et al. [43] studied 18 Au and Ag nanoparticles with different morphologies and assemblies, such as ball, rod [44], flower [45], star [46], core/shell [47], hollow [48], octahedron [49], core/satellite [50], and chain-like aggregate (Figure 3). They quantitatively compared their SERS performance in pH detection and found that the single Ag chain, Ag core/satellites, Au coated Ag core/satellites, and Au core/satellites nanoassembly showed more efficient pH SERS sensing than others, which is consistent with the results of the hot spot effect described earlier.

Among many designs of pH Raman probes, the traditional pH SERS probes combine pH sensitive molecules with single particles through chemical or physical action, which is a relatively simple design. It has the advantages of small volume and low difficulty in preparation. In an earlier study, Kneipp et al. [51] demonstrated mobile SERS nanosensors made of gold nanospheres’ aggregates and 4-mercaptobenzoic acid (pMBA) for monitoring local pH changes in living NIH/3T3 cells. As more and more research groups used precious metal nanoparticles functionalized by acid ligands to carry out surface enhanced Raman scattering to realize the highly sensitive monitoring of intracellular pH value, researchers gradually turned their attention to the dependence of pH SERS sensitivity on the morphology of nanoparticles (Figure 4a). Schwartzberg et al. [52] realized the SERS measuring scale for hollow nanostructures for the first time and demonstrated it as a pH sensor model system. Compared with the previous pH SERS probes, its resolution was nearly doubled, and its accuracy was improved by 10 times (Figure 4b). This new detection platform is an important step forward in potential biosensor applications. It is well known that the apparent dissociation constant (pKa) of acid ligands (pH Raman reporter molecules) is sensitive to nanoparticle curvature, and that flower-shaped or star-shaped nanoparticles have higher curvature than near-spherical nanoparticles. In addition, the phenomenon that flower-like and stellate-like nanoparticles have single-particle Raman signals has been widely reported. In this connection, Zhang [53] successfully obtained the visualized intracellular pH value (pHi) and the pHi curve of living cells through the functionalized nanostars, thus visualizing the dynamic distribution of pHi in a single cell (Figure 4c). Similarly, Zhang’s research group [54] compared the pKa information behavior and pH SERS sensitivity of near-spherical isotropic and flower-like anisotropic gold coated silver nanoparticles (Figure 4d). The experimental results show that the flower-like nanoparticles with higher curvature exhibit a narrower pH sensitivity range and a higher sensitivity to pH range than the near-spherical nanoparticles. As we know, the sensitivity of SERS signal of pH probe was not only affected by the nanoparticles, but also the composition of nanoparticles.

Traditional single particle pH sensors based on surface enhanced Raman scattering often rely on the physical enhancement effect generated by the aggregation of nanoparticles, but the aggregation degree of nanoparticles is difficult to be accurately controlled, which leads to poor repeatability of SERS signals of these kinds of pH sensors. Therefore, many research groups designed a series of array platforms for the construction of pH SERS probes with good repeatability and stability on this basis. Dzięcielewski et al. [55] based on chemical etching of GaN crystals developed a substrate similar to “nanowhiskers” array for the construction of pH SERS probe (Figure 4e). However, the array structure has an obvious disadvantage. Compared with the single particle pH SERS probe, its size is often much larger, which makes it an obstacle to enter the cell to detect pH. Similarly, Zhao’s group [56] successfully developed a pH sensor using carbon nanotubes modified with bimetallic nanoparticles (Figure 4f). A carbon nanotube can continuously measure pH values in its vicinity over a range of 5.6–8.2 pH units. As we all know, the size of the dimer is often much smaller than that of the array structure, so on this basis, Pallaoro et al. [57] used hexanediamine (HMD) to connect two silver nanospheres functionalized by pH-sensitive molecule 4-mercaptobenzoic acid together. The addition of the protective agent polyvinylpyrrolidone (PVP) could prevent the further polymerization of nanoparticles well, which made the SERS probe of dimer pH successfully prepared (Figure 4g). Chen et al. [58] systematically studied the formation and evolution of dynamic surface enhanced Raman scattering hot spot, and on this basis, they prepared gold nanoparticles supported pH responsive poly (2-vinylpyridine) composite microgel through in situ chemical reduction method (Figure 4h). What is interesting about the pH response mechanism of such particles is that when the pH value of the external environment was from 2.0 to 5.0, the nanometer gap between AuNPs could be easily adjusted, creating a huge SERS hot spot for electromagnetic enhancement.

Traditional pH SERS probes tend to generate different degrees of aggregation in high ionic strength medium and complex biological system, and the probe molecules on it may also interact with biological molecules. As a result, SERS detection results were not reliable enough and biocompatibility was low, which made it difficult for traditional pH SERS probes to be used for intracellular pH detection. In order to overcome this interference, the strategy often used by researchers is to modify a layer of BSA or other substances on the surface of the designed pH SERS probes to increase the biocompatibility and stability of the pH probes, so that the pH SERS probes can be used for cell pH monitoring accurately and sensitively. Based on this, Zheng’s research group [59] also prepared 4-mercaptopyridine functionalized gold nanospheres modified with bovine serum protein as pH SERS probe (Figure 4i), which was used to monitor the pH distribution of living cells. It was found that the nanoparticles modified with bovine serum protein had high sensitivity to pH value, good biocompatibility and stability. Bi et al. [60] prepared a gold nanoarray platform with good SERS sensitivity by adopting the efficient evaporation self-assembly method, and on this basis developed a pH sensor based on SERS with high sensitivity bovine serum protein coated 4-mercaptopyridine connected gold nanorod array (Figure 4j). It was successfully used to detect pH in the blood of mice, providing a favorable platform for the development of a sensitive bioon chip platform designed for the point of care device. In a different way, Wang [61] encapsulated the constructed silver nano pH SERS probe in the silicon layer (Figure 4k), whose small hole could block the large biomolecules outside the sensor without preventing H+ from entering into it and interacting with 4-mercaptobenzoic acid on the silver surface. The stability, pH sensitivity and reliability of the probe are maintained. Bi et al. [62] developed a simple, sensitive and stable Prussian blue (PB) cage pH response SERS probe (Figure 4l), which was used to quantify the dynamic change of pH (pHi) in the living cells. Compared with other pH SERS probes, the subnanometer porous PB shells in the structure could allow the free diffusion reaction of H^+^ or OH^−^ while blocking the contact between proteins, DNA and other biological sulfhydryl compounds in the cell. In addition, the C-N triple bond group of nitrile in PB shell can be used as an internal standard without background interference to accurately analyze the distribution of probes in the whole cell. Table 1 is a summary of some pH SERS probe designs reported above.

## 5. pH SERS Probe Was Used for Cell pH Value Monitoring and pH Imaging

The pH near the cell surface is called extracellular pH (pHe). Extracellular pH (pHe) and intracellular pH (pHi) are not only the important regulatory factors affecting a variety of physiological activities of cells (such as cell proliferation, differentiation and apoptosis, ion transport, endocytosis, muscle contraction, nucleic acid formation and metabolite release). It is also one of the important parameters for the study of related pathological processes, so the acids and bases inside and outside the cell play an important role in chemical reactions and biological processes [63,64]. Abnormal pHe values are associated with various pathological states, such as tumor, iron-deficiency stroke, infection, and inflammation [65]. Similarly, small changes in pHi can lead to major changes in metabolism that can lead to disease. It is worth noting that extracellular pH (pHe) is often used as an important marker of cancer. This is mainly due to the Warburg effect (normally differentiated cells rely mainly on oxidative phosphorylation of mitochondria for cell energy, while most tumor cells rely on aerobic glycolysis) and the action of carbonic anhydrase on the cell surface, tumor cells will produce a large amount of extracellular acid, which leads to a slightly lower pHe (6.2–6.9) of tumor cells than pHi (7.2–7.4) [66,67] (Figure 5). In addition to cancer, there are other diseases that are closely related to pHe and pHi abnormalities in cells, such as quite a number of neurological diseases [68]. Due to the abnormal cell pH (pHe and pHi) in these diseases, the phenomenon of abnormal cell pH has attracted extensive attention in clinical medicine and researchers. Therefore, accurate and sensitive monitoring and intuitive imaging characterization of pHe and pHi are very worthy of expectation for the pathological research and the development of new treatment methods of these diseases.

The perception of pH value of living cells is of great significance for understanding various physiological and pathological processes [69,70]. As an ultrasensitive and nondestructive spectroscopic technology, surface enhanced Raman spectroscopy has been widely used in the monitoring and imaging studies of cell pH value [71,72]. However, there are still three major problems to be overcome when applying pH SERS probe to cell pH value detection and imaging: (1) in the cellular system, the selected peak used to detect the pH response of the overall buffer solution is still reliable; (2) because the aggregation trend of nanoparticles after being absorbed by cells would increase [73], which would affect the reliability of pH SERS probe monitoring results; (3) whether a variety of biomolecules existing in the living cell system can replace or interact with Raman probes or probe molecules, and whether pH SERS probe can remain intact, will lead to changes in the stability and reliability of probes [59]. Sun et al. [33] immobilized 4-MBA on Au Quasi-3D plasma nanostructure array (Q3D-PNA) through the chemical action of Au-S bond, and constructed a pH probe based on surface enhanced Raman spectroscopy, which has high sensitivity and repeatability. It was successfully applied to draw pHe map of living cells (Figure 6a). Xu et al. [34] used 4-mercolbenzoic acid functionalized gold nanoparticle array as substrate for pH imaging of three different types of cells. Figure 6b is the SERS image of pH within 3 h after adding apoptosis inducer to SCC-4 cell. Guo et al. [74] developed a nano-microtube with size less than 200 nm and SERS activity for detecting intracellular pH value (pHi) of a single eukaryotic cell. Li et al. [75] prepared a simple plasma Raman probe using nanostars as a Raman enhanced substrate, 4-Mpy as a Raman reporter molecule, and bovine serum albumin as a protective molecule. Subsequently, SERS imaging technology was used to monitor the changes of lysosomal pH in the process of autophagy and apoptosis. According to SERS imaging, the pH value of each pixel was given, so that the pHi distribution map could be drawn (Figure 6c).

## 6. pH SERS Probe Combined with Other Technologies

In numerous pH imaging, fluorescence studies are widely used in cell pH monitoring imaging, because its cost is relatively low and good security in in vitro research [76,77]. However, in cell pH detection and imaging, fluorescence spectrum technology has defects such as easy quenching, photobleaching, large background interference and slow imaging, while SERS spectrum technology did not have these problems. Based on this, Yang et al. [78] constructed a nanoprobe with dual signals of fluorescence and surface enhanced Raman scattering in recent years to sense the pH value in cells (Figure 7a). In this study, they designed a double-signal core-shell nanoprobe with fluorescence and SERS by co-functionalization of SiO_2_-coated gold nanorods by using fluorescein isothiocyanate (FITC) and p-mercaptobenzoic acid (pMBA). Then, through the study and analysis of the fluorescence and SERS spectra of the nanoprobes at different pH values, the double-signal response to the intracellular pH value could be realized. Different from the former, Yue et al. [79] prepared a new generation of fluorescence-SERS dual-mode pH sensors by embedding functionalized nanoparticles into stimulus-responding hydrogels. Such sensors were equipped with synergistic enhancement of the properties of each component: improving the mechanical strength of hydrogels and reducing nanoparticle aggregation (Figure 7b). The complementary advantages of fluorescence and SERS in response to pH made it successfully applied to study the changes of pH value in microenvironment caused by external stimulation of different cells. As shown in Figure 7d, Yue’s research group [80] also successfully developed a nanosensor based on fluorescence imaging and SERS spectrum in recent two years to monitor the dynamic changes of pH in mitochondrial microenvironment of two kinds of cancer cells (HepG2 and MCF-7 cells) and one kind of normal cells (LO2 cells) during the process of photodynamic therapy (PDT). This study also provides a reference for evaluating the side effects of PDT.

In addition to enhancing the reliability of pH sensors by combining with fluorescence spectroscopy technology, simple, convenient, fast and sensitive electrochemistry [81] has also gradually entered the field of view of researchers. Pan’s research group [82] designed a surface enhanced Raman scattering active microneedle to detect redox potential and pH value at rat joints. The design of composite SERS probe was based on the microneedles used in traditional Chinese medicine acupuncture and moxibustion. The grooves formed by chemical etching on the microneedles contain two kinds of molecules, namely redox sensitive and pH sensitive, respectively, and then enter the muscle at the minimum cost of damage, making the muscle undergo a redox state and pH dynamic evolution for 5 min (Figure 7c). This has advanced the pathological study of diseases such as arthritis and will also be a multifunctional analytical tool to advance biomedical research. Many research groups have made many attempts and efforts to combine pH detection with biological diagnosis and treatment. As shown in Figure 7e, Jung et al. [83] designed a nanoprobe with both pH response and photothermal therapy. Multi-function pH probe in the golden ball surface is designed to develop a positive charge and negative charge. Under the action of weak acid, they are induced to gather together rapidly by electrostatic interaction, while the emergence of coupled plasma model causes their absorption to transfer to the near-infrared region, where a direct selective absorption transfer to depth is carried out, through tissue and heat treatment. Figure 7 is a schematic of the examples mentioned above.

## 7. Conclusions

High sensitivity of SERS, excellent multiplexing ability of a single excitation source, low background and minimal photobleaching of SERS nanotags lead to its application in cell pH detection imaging technology becoming more and more widespread. At present, the main problems of developing pH SERS probes that can be used for cell pH detection and imaging focus on ensuring that they can maintain good detection ability and avoid the interference of microenvironment, which requires researchers to design and construct well-defined, uniform and stable metallic nanostructures and assemblies. In addition, the synthesis and discovery of pH-sensitive Raman reporter molecules with highly stable and repeatable signals could be an important step forward in this field. In recent years, the pH SERS probes reported tended to have a wide pH response range, so the subsequent pH SERS probes should develop towards a narrower response range and higher sensitivity, so that they could be better applied in biological systems. In addition, it is also very important to further develop highly selective and non-toxic probe structures. We believe that the additional complementary information obtained by combining with other different independent methods can improve the reliability of probe signals or become a multi-functional intelligent probe so that pH SERS probes can be better applied in the biomedical field.

By summarizing and classifying the latest and most representative research reports on pH SERS probes, this paper mainly reviewed the design and assembly strategies of several major types of pH probes and their structures in this field, as well as the latest progress in pH imaging in and out of cells. At present, the purpose of the research groups to study pH SERS probe is mainly to make the signal stable and reliable, and to make the imaging fast and large. In addition, in the construction of probes for cell pH monitoring and SERS imaging, the design of protective matrix and the enhancement of biocompatibility of nanometer tags also need the efforts of future researchers. Finally, Table 2 is a summary of some of the full abbreviations that appear in this article.

## Figures and Tables

**Figure 1 biosensors-11-00282-f001:**
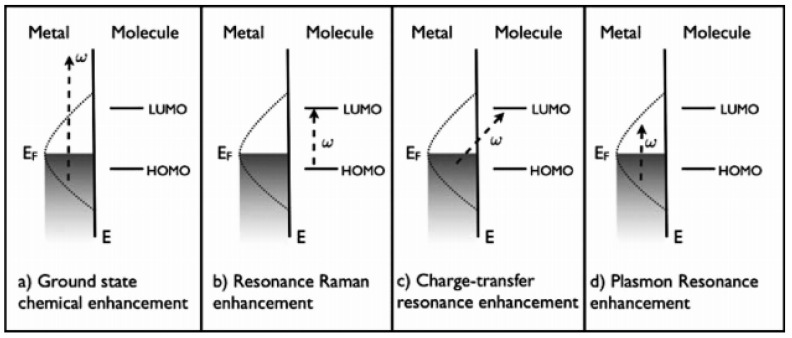
Illustration of the different types of enhancement mechanism. Reprinted with permission from ref. [10]. Copyright 2008 Royal Society of Chemistry. Chemical enhancement mechanism: (**a**) Diagram of the mechanism of ground state chemical enhancement. (**b**) Diagram of the mechanism of resonance Raman enhancement. (**c**) Diagram of the mechanism of Charge-trancsfer enhancement. (**d**) Diagram of the mechanism of plasmon resonance enhancement.

**Figure 2 biosensors-11-00282-f002:**
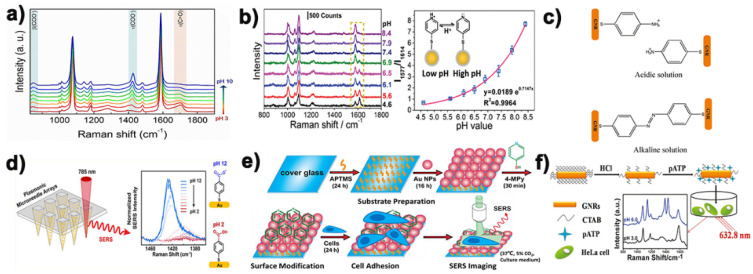
(**a**) Representative SERS spectrum of 4-MBA-AuNPs exposed to solutions at different pH values. Reprinted with permission from ref. [38]. Copyright 2019 Capocefalo, Mammucari, Brasili, Fasolato, Bordi, Postorino and Domenici. (**b**) Normalized SERS spectrum and pH standard curve of 4-Mpy functionalized pH nanosensors and changes of 4-Mpy molecular structure. Reprinted with permission from ref. [27]. Copyright 2019 Royal Society of Chemistry. (**c**) Schematic diagram of 4-ATP-GNRs structure in acidic and alkaline solutions. Reprinted with permission from ref. [19]. Copyright 2011 American Chemical Society. (**d**) 4-MBA modified plasma microneedle array pH response mechanism diagram. Reprinted with permission from ref. [25]. Copyright 2019 American Chemical Society. (**e**) Schematic diagram of 4-Mpy modified nanospheres array cell pH probe. Reprinted with permission from ref. [34]. Copyright 2018 American Chemical Society. (**f**) Flow chart of 4-ATP gold nanorods pH sensor preparation. Reprinted with permission from ref. [19]. Copyright 2011 American Chemical Society.

**Figure 3 biosensors-11-00282-f003:**
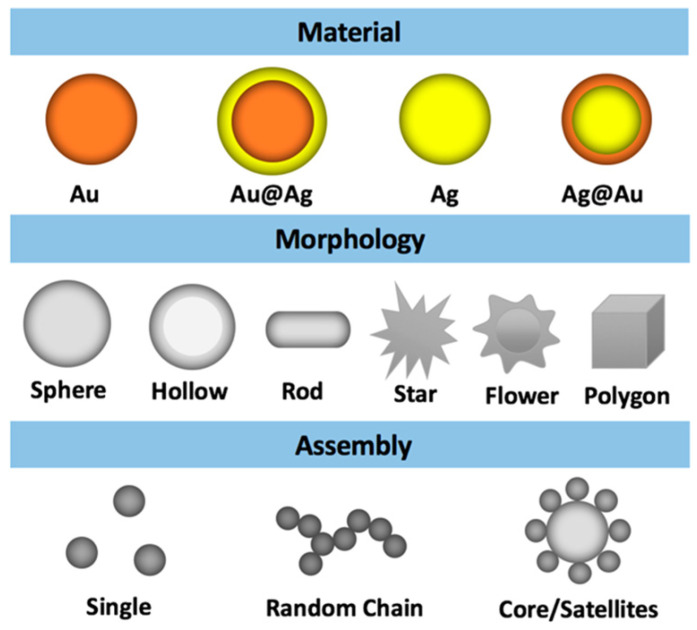
Types of Au and Ag nanoparticles with different morphologies and assembly states. Reprinted with permission from ref. [43]. Copyright 2019 American Chemical Society.

**Figure 4 biosensors-11-00282-f004:**
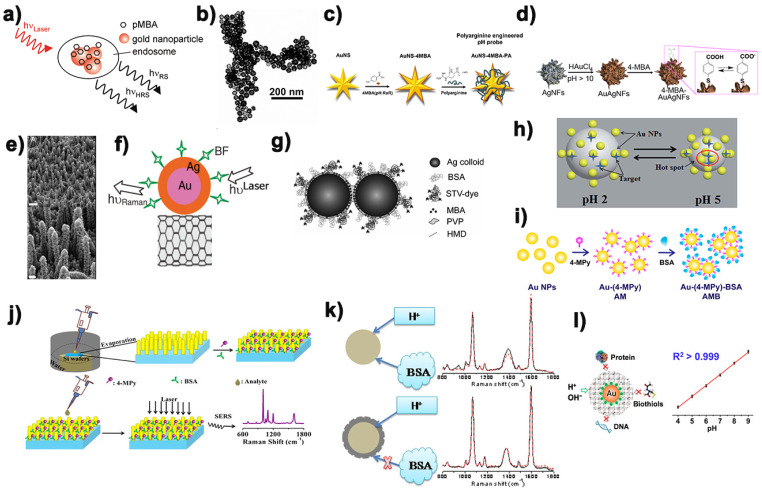
Some examples of traditional single particles pH SERS probes (top panel), assembled pH SERS probe (middle panel) and encapsulated pH SERS probe (bottom panel). (**a**) Schematic diagram of pMBA modified gold nanospheres pH SERS probe. Reprinted with permission from ref. [51]. Copyright 2010 American Chemical Society. (**b**) TEM of hollow gold nanospheres. Reprinted with permission from ref. [52]. Copyright 2006 American Chemical Society. (**c**) Schematic diagram of preparing gold nanostar pH SERS probe. Reprinted with permission from ref. [53]. Copyright 2020 American Chemical Society. (**d**) Schematic diagram of preparing Au coated Ag nanoflowers pH SERS probe. Reprinted with permission from ref. [54]. Copyright 2020 American Chemical Society. (**e**) TEM of Au-sputtered GaN nanowhiskers array. Reprinted with permission from ref. [55]. Copyright 2020 Elsevier B.V. (**f**) Schematic diagram of Ag Au bimetallic nanoparticles modified carbon nanotubes. Reprinted with permission from ref. [56]. Copyright 2008 Royal Society of Chemistry. (**g**) Schematic diagram of silver nanospheres dimer. Reprinted with permission from ref. [57]. Copyright 2010 John Wiley & Sons, Inc. (**h**) Schematic diagram of gold nanospheres supported poly (2-vinylpyridine) composite microgel. Reprinted with permission from ref. [58]. Copyright 2017 Royal Society of Chemistry. (**i**) Schematic diagram of bovine serum protein modified 4-Mpy functionalized gold nanorods. Reprinted with permission from ref. [59]. Copyright 2014 American Chemical Society. (**j**) Schematic diagram of serum protein coated 4-Mpy connected gold nanorod array. Reprinted with permission from ref. [60]. Copyright 2018 American Chemical Society. (**k**) Schematic diagram of silicon encapsulated silver nanoparticles with stable pH response. Reprinted with permission from ref. [61]. Copyright 2012 American Chemical Society. (**l**) Schematic diagram of Prussian blue (PB) cage gold-coated nanorods with stable pH response. Reprinted with permission from ref. [62]. Copyright 2020 American Chemical Society.

**Figure 5 biosensors-11-00282-f005:**
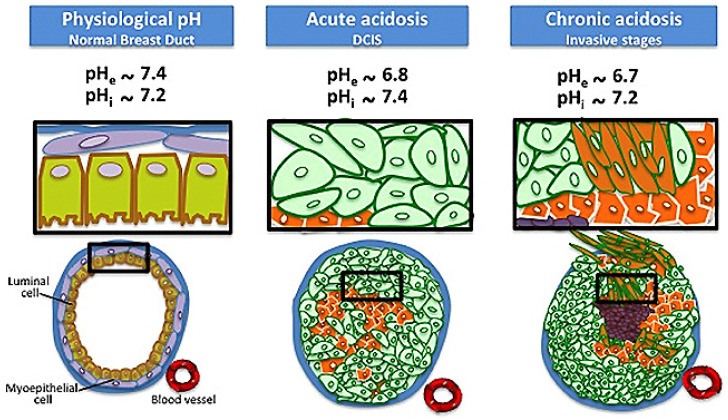
Schematic of reversal changes in extracellular and intracellular pH values of cancer cells compared with normal cells. Reprinted with permission from ref. [66]. Copyright 2013 Damaghi, Wojtkowiak and Gillies.

**Figure 6 biosensors-11-00282-f006:**
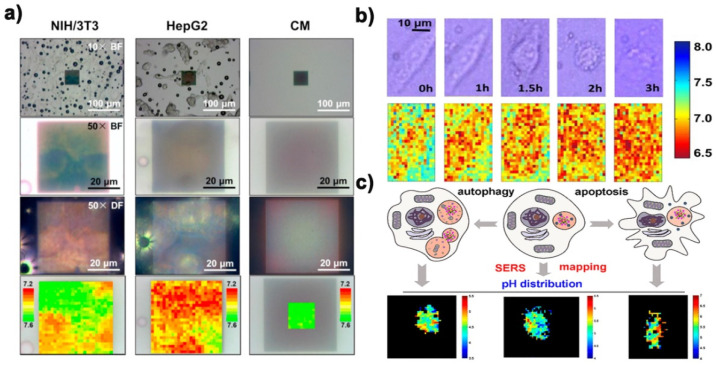
(**a**) Optical images and pHe SERS map of NIH/3T3 and HepG2 cells after culture on 4-MBA modified Q3D-PNA SERS substrates. Reprinted with permission from ref. [33]. Copyright 2015 Elsevier B.V. (**b**) Bright field microscope image (upper panel) and real-time in situ pHe images of SCC-4 cell (lower panel). Reprinted with permission from ref. [34]. Copyright 2018 American Chemical Society. (**c**) In situ SERS imaging and schematic of pHi during autophagy and apoptosis of lysosomes. Reprinted with permission from ref. [75]. Copyright 2019 American Chemical Society.

**Figure 7 biosensors-11-00282-f007:**
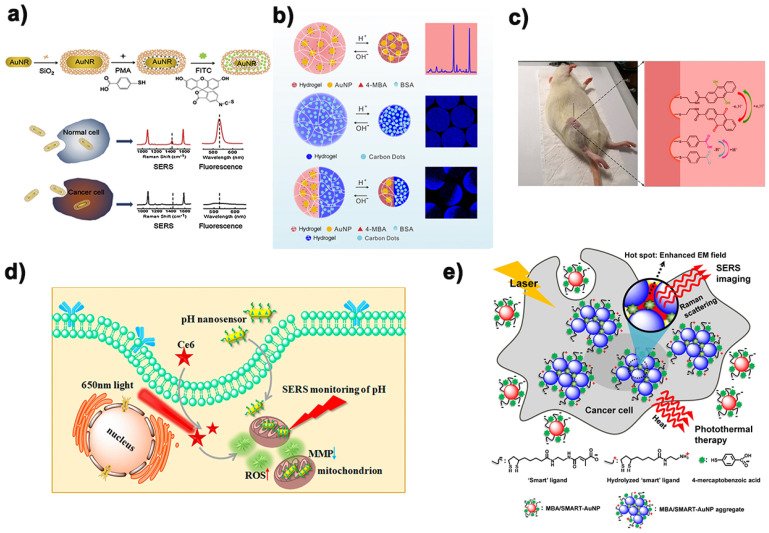
Some examples of pH SERS probe combined with other technologies. (**a**) SiO_2_ and PMA-FITC coated AuNRs nanoprobe preparation and its mechanism in cells. Reprinted with permission from ref. [78]. Copyright 2018 Elsevier B.V. (**b**) Schematic diagram of pH-induced swell and shrinking of single-phase hydrogel microparticles encapsulating AuNPs-(4MBA)-BSA (upper), single-phase hydrogel microparticles encapsulating carbon dots (middle), and Janus hydrogel microparticles encapsulating AuNPs-(4MBA)-BSA and carbon dots, respectively, in duals semispheres (lower). Reprinted with permission from ref. [79]. Copyright 2017 American Chemical Society. (**c**) Schematic diagram of multiplexing SERS active microneedles for simultaneous measurement of redox potential and pH value. Reprinted with permission from ref. [82]. Copyright 2019 American Chemical Society. (**d**) Procedure of the selective determinations of microenvironmental changes in mitochondria during PDT progress. Reprinted with permission from ref. [80]. Copyright 2020 American Chemical Society. (**e**) Schematic diagram of the working mechanism of 4-MBA-functionalized smart nanoparticles in cancer cells. Reprinted with permission from ref. [83]. Copyright 2013 American Chemical Society.

**Table 1 biosensors-11-00282-t001:** Summary of some pH SERS probes.

SERS Substrate	Reporter of pH	Range of pH	Target	Ref.
Gold nanospheres	4-mercaptobenzoic acid	5.0–6.9	NIH/3T3 cells.	[51]
Hollow gold nanospheres	4-mercaptobenzoic acid	3.5–9.0	——	[52]
Gold nanostars	4-mercaptobenzoic acid	3.2–9.4	MCF7 cells	[53]
Gold-coated silver nanoparticles and nanaflowers	4-mercaptobenzoic acid	5.0–8.0	A549 cells	[54]
Au-sputtered GaN nanowhiskers	4-mercaptopyridine	2.23–12.35	——	[55]
Bimetallic-nanoparticle-decorated carbon nanotubes	Biotin-fluorescein moleules	5.6–8.2	Bimetallic-nanoparticle-decorated carbon nanotubes	[56]
Silver nanospheres dimer	4-mercaptobenzoic acid	3.2–9.0	HeLa cells	[57]
Gold nanoparticles/poly(2-vinylpyridine) microgels	4-mercaptobenzoic acid	2.0–5.0	——	[58]
Gold nanoparticles coated with bovine serum protein	4-mercaptopyridine	4.0–9.0	CaSki cells	[59]
Gold nanorods array platform	4-mercaptopyridine	3.0–8.0	Mouse boold	[60]
Silver nanoparticles coated with silica layer	4-aminobenzene mercaptan	2.0–10.0	Macrophage cells	[61]
gold nanoparticles coated with Prussian blue cage	4-mercaptopyridine	1.69–11.2	Adenocarcinoma cells	[62]

**Table 2 biosensors-11-00282-t002:** Abbreviations used in the text.

Abbreviations	Full Name	Abbreviations	Full Name
AuNPs	Gold nanoparticles	AuNRs	Gold nanorods
SERS	Surface enhanced Raman scattering	Hpyt	5-(4-pyridyl)-1,3,4-oxadiazole-2-thiol
ER	Endoplasmic reticulum	LSPR	Local surface plasmonic resonance
EM	Electromagnetic enhancement mechanism	HMD	Hexanediamine
CM	Chemical enhancement mechanism	PVP	Polyvinylpyrrolidone
4-MBA	4-mercaptobenzoic acid	PB	Prussian blue
4-Mpy	4-mercaptopyridine	pHe	Extracellular pH
4-ATP	4-aminobenzene mercaptan	pHi	Intracellular pH
2-ABT	2-mercaptobenzoic acid	Q3D-PNA	Au Quasi-3D plasma nanostructure array
TNA	Thionicotinamide	PDT	Photodynamic therapy
iTNA	Thioisonicotinamide	BSA	Bovine serum protein

## Data Availability

Not applicable.

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
