# Peer review of "Recent Progress of SERS Nanoprobe for pH Detecting and Its Application in Biological Imaging"

_biosensors, 2021, doi:10.3390/bios11080282_

Round 1

Reviewer 1 Report

Comments to the author:

In this review the authors reported the progress made on the design of SERS probes for pH sensing and imaging. The review is very didactic, starting from the basis of SERS spectroscopy and the importance of pH sensing. The authors summarize and explained the mechanism of action of the main Raman pH sensitive probes and the best examples in the field. Besides they also included a section about the combination of pH imaging with other techniques. Overall, this contribution presents an excellent overview of how SERS spectroscopy has been used for pH sensing and imaging. I would recommend the publication of this contribution and I added just a few minor issues should that be addressed.

The following are some questions and suggestions for improving their work:

Some acronyms (e.g. SERS) are defined several times along the text.

Plasma nanoparticles should be replaced by plasmonic nanoparticles.

Page 2, line 80 the authors should rephrase : ‘nondestructive and non-destructive photon’.

Could the authors indicate the pH range of 4-mercaptopyridine (4-Mpy) as they did for the other main two pH sensitive probes?

As a final suggestion a table including the different parameters (e.g. type of NP, sensing probe, pH range…etc) would be very useful for the readers.

Author Response

Thank you for your comments concerning our manuscript entitled "Recent progress of SERS Nanoprobe for pH Detecting and its Application in Biological Imaging". Those comments are valuable and very helpful. We have read through comments carefully and have made corrections. The responses to the reviewer's comments are marked in yellow.

The following is a step-by step explanation of your comments.

Point 1: Some acronyms (e.g. SERS) are defined several times along the text.

Response 1: We have removed the acronyms defined several times in the manuscript upload this time.

Point 2: Plasma nanoparticles should be replaced by plasmonic nanoparticles.

Response 2: We have replaced plasma nanoparticles with plasmonic nanoparticles (Page2, line 72).

Point 3: Page 2, line 80 the authors should rephrase : ‘nondestructive and non-destructive photon’.

Response 3: We have rephrased the "nondestructive and non-destructive photon" (Page2 line80).

Point 4: Could the authors indicate the pH range of 4-mercaptopyridine (4-Mpy) as they did for the other main two pH sensitive probes?

Response 4: The pH range of 4-mercaptopyridine (4-Mpy) could be found in Table 1 and page 6, line 206.

Point 5: As a final suggestion a table including the different parameters (e.g. type of NP, sensing probe, pH range…etc) would be very useful for the readers.

Response 5: We have added Table 1 to the manuscript to summarize the different paraments of some pH SERS probes.

Reviewer 2 Report

The manuscript entitled “Recent progress of SERS nanoprobe for pH detecting and its application in biological imaging” by Zhang et al. presents a review on Raman reporters/SERS substrates used for pH SERS detection/SERS imaging on biological molecular probes of interest, namely cells. The manuscript is well structured, and the data is well-reviewed with many examples; however, section 2.1, “SERS”, is poor and lack information essential to understand the following sections.

Thus, I do not recommend publication after major revisions.

Comments:

  1. Figures 1, 3 and 6 are tiny and with low resolution. The captions in each image are very difficult to read, even in pdf format.
  2. Page 1, line 37: It should be referencing a review article about the subject
  3. Page 2, line 40: What do the authors mean with good reuse ability? Normally the SERS substrates, namely metal colloids, cannot be reuse. Can the authors explain this?
  4. Section 2.1: this section needs more references support (please see: Phys.Chem.Chem.Phys., Nat Rev Mater 1, 16021 (2016); http://dx.doi.org/10.5772/66097, Nat Rev Chem 1, 0060 (2017)).

The authors need to explain or talk about metallic nanoparticles and the influence of morphology, size and nature on the SERs signal (explain the localized surface plasmon resonance). This is very important to understand section 4.

Explain what hotspots are (they mention this below in section 4 and Figure 3)

Explain what SERS imaging is for the readers to understand the importance of developing such pH SERS sensors for biological SERs imaging (See: ).

It was better to have a figure describing the EM and CM (see: Surface-enhanced Raman scattering sensors based on hybrid nanoparticles, in Microsensors, Intech, 2011, ch. 7; Chem. Soc. Rev., 2008, 37, 1061–1073.)

Reference 10 (line 88) does not belong there. Please revised.

  1. Section 2.2, line 106, Reference 18 does not belong there. Please revised.
  2. The authors report some works from the literature in section 3 for 4MBA (21-23), 4-Mpy (24-26), and 4-ATP (27,28). Why are these papers not cited in the following sections for each Raman reporter? Please add these references in each sub-section (3.2.1, 3.2.2 and 3.2.3)
  3. In section 3.2. the authors could make a table summarizing the pH Raman reported used in the several works mention the main text, the SERS substrate and target (e.g. cells, tumour, etc.)
  4. Page 6, line 233-237: The authors claimed “In this study, Zhang et al. [44] studied 18 Au and Ag nanoparticles with different morphologies and assemblies, such as ball, rod [46], flower [47], star [48], core/shell [49], hollow [50], octahedron [51], core/satellite [52], and chain-like aggregate (Figure 2). And quantitatively compared their SERS performance in pH detection.” What were the conclusions? The Raman reported used? This is incomplete. Please revise
  5. Page 7, line 320: What is live fine?
  6. What is the Warburg effect? Please explain this in the manuscript.
  7. Figure 5b only demonstrate 1 type of cell, and the authors claim in the text that the work was performed in 3 different types of cells. Please revised. What means the time (h) in the optical image?
  8. Page 9, line 387: What is SMB?
  9. In figure 6, The image (c) correspondent to ref 82; however, in the main text, the authors report the work of ref 82 to Figure 6d. Please revise
  10. Page 9, line 423: What is PDT?
  11. The authors could make an abbreviation list for support.

Other comments:

  1. Section 2, is not SRES but SERS (page 2, line 68)
  2. Page 4, line 156: I believe that it is 4-mercaptobenzoic acid instead of 4-mercurybenzoic acid. Please revise
  3. The authors, in some cases, use pH SERS sensors and other times use SERS pH sensors. Please always use the same terminology
  4. For intracellular pH values, the authors use pHi and i-pH 8line 261, page 6). Please use only one terminology.
  5. Page 6, line 262: it is nanostar instead of nanosar
  6. Page 10, line 443: I believe that is Figure 6e instead of Figure 6d. Please revise

Author Response

We appreciate the time and effort that you dedicated to providing feedback on our manuscript and are grateful for the insightful comments on and valuable improvements to our paper. We have incorporated most of the suggestions made by the reviewers. Those change changes  are highlighted in yellow within the manuscript.

The following is a step-by step explanation of your comments.

Point 1: Figures 1, 3 and 6 are tiny and with low resolution. The captions in each image are very difficult to read, even in pdf format.

Response 1: We have changed the small image in the article to a higher resolution one and made some typesetting and size changes (FIgure2, 4, 7).

Point 2: Page 1, line 37: It should be referencing a review article about the subject.

Response 2: Page 1, line 37: We have replaced reference 1 with a review article on the subject.

Point 3: Page 2, line 40: What do the authors mean with good reuse ability? Normally the SERS substrates, namely metal colloids, cannot be reuse. Can the authors explain this?

 Response 3: The good reuse ability mainly means that SERS has strong feature detection ability, which can be used for repeated detection.

Point 4: Section 2.1: this section needs more references support (please see: Phys.Chem.Chem.Phys., Nat Rev Mater 1, 16021 (2016); http://dx.doi.org/10.5772/66097, Nat Rev Chem 1, 0060 (2017)).

The authors need to explain or talk about metallic nanoparticles and the influence of morphology, size and nature on the SERs signal (explain the localized surface plasmon resonance). This is very important to understand section 4.

Explain what hotspots are (they mention this below in section 4 and Figure 3)

Explain what SERS imaging is for the readers to understand the importance of developing such pH SERS sensors for biological SERs imaging (See: ).

It was better to have a figure describing the EM and CM (see: Surface-enhanced Raman scattering sensors based on hybrid nanoparticles, in Microsensors, Intech, 2011, ch. 7; Chem. Soc. Rev., 2008, 37, 1061–1073.)

Reference 10 (line 88) does not belong there. Please revised.

 Response 4: At the end of Section 2.1, we have added some supplementary instructions suggested by reviewers in the highlighted yellow section.

Point 5: Section 2.2, line 106, Reference 18 does not belong there. Please revised.

Response 5: Page 3, line 127: Reference 20 has been revised.

Point 6: The authors report some works from the literature in section 3 for 4MBA (21-23), 4-Mpy (24-26), and 4-ATP (27,28). Why are these papers not cited in the following sections for each Raman reporter? Please add these references in each sub-section (3.2.1, 3.2.2 and 3.2.3)

Response 6: We have added references to the three ph-responsive molecules in sections 3.2.1, 3.2.2, and 3.2.3, and have made some adjustments to the references in the previous sections .

Point 7: In section 3.2. the authors could make a table summarizing the pH Raman reported used in the several works mention the main text, the SERS substrate and target (e.g. cells, tumour, etc.)

Response 7: At the end of Section 3.2, we added the Table 1 summarizing the parameters of pH SERS probes.

Point 8: Page 6, line 233-237: The authors claimed “In this study, Zhang et al. [44] studied 18 Au and Ag nanoparticles with different morphologies and assemblies, such as ball, rod [46], flower [47], star [48], core/shell [49], hollow [50], octahedron [51], core/satellite [52], and chain-like aggregate (Figure 2). And quantitatively compared their SERS performance in pH detection.” What were the conclusions? The Raman reported used? This is incomplete. Please revise

Response 8: Page 7, line 265-269: The conclusions of Zhang et al. 's study have been added and the description error has been corrected.

Point 9: Page 7, line 320: What is live fine?

Response 9: Page 9. line 351: We are so sorry for that the description of live fine is wrong, and we have corrected it with living cells.

Point 10: What is the Warburg effect? Please explain this in the manuscript.

Response 10: Page 11, line383-385: The explanation of Warburg effect has been added.

Point 11: Figure 5b only demonstrate 1 type of cell, and the authors claim in the text that the work was performed in 3 different types of cells. Please revised. What means the time (h) in the optical image?

Response 11: Page 12, line 416-414: A description of Figure 6b has been added. Tthe time (h) in the optical image means the time after adding apoptosis inducer to SCC-4 cell.

Point 12: Page 9, line 387: What is SMB?

Response 12: Page 12, line 420: The SMB means the pH SERS probe constructed using nano star as Raman enhanced substrate, 4-MPY as Raman reporter molecule and bovine serum albumin as protective molecule. We decided to remove it because we felt it was not necessary and would give readers a bad reading experience.

Point 13: In figure 6, The image (c) correspondent to ref 82; however, in the main text, the authors report the work of ref 82 to Figure 6d. Please revise

Response 13: Page 14, line 482: We have revised the references in Figure 7d.

Point 14: Page 9, line 423: What is PDT?

Response 14: Page 13, line 455: PDT stands for photodynamic therapy and we have added an explanation of this in the article.

Point 15: The authors could make an abbreviation list for support.

Response 15: The abbreviation list has been added in the ending of this manuscript (Table 2).

Point 16: Section 2, is not SRES but SERS (page 2, line 68)

Response 16: Page 2, line 68: SRES has been corrected with SERS.

Point 17: Page 4, line 156: I believe that it is 4-mercaptobenzoic acid instead of 4-mercurybenzoic acid. Please revise

Response 17: Page 4, line 177: We have replaced 4-mercurybenzoic acid with 4-mercaptobenzoic acid.

Point 18: The authors, in some cases, use pH SERS sensors and other times use SERS pH sensors. Please always use the same terminology

Response 18: We have unified the description of the whole text as pH SERS sensors.

Point 19: For intracellular pH values, the authors use pHi and i-pH 8line 261, page 6). Please use only one terminology.

Response 19: We have unified the description of the whole text as pH SERS (Page 2, line 52; line 296Page 5, line 196; Page 7, line 282, line 296).

Point 20: Page 6, line 262: it is nanostar instead of nanosar

Response 20: Page 7, line 294: The word nanosar has been corrected with nanostar.

Point 21: Page 10, line 443: I believe that is Figure 6e instead of Figure 6d. Please revise

Response 21: Page 13, line 475: We have replaced Figure 6d with 4-Figure 6e.